# AFDGCF: ADAPTIVE FEATURE DE-CORRELATION GRAPH COLLABORATIVE FILTERING FOR RECOMMENDATIONS

## ABSTRACT

Collaborative filtering methods based on graph neural networks (GNNs) have witnessed significant success in recommender systems (RS), capitalizing on their ability to capture collaborative signals within intricate user-item relationships via message-passing mechanisms. However, these GNN-based RS inadvertently introduce a linear correlation between user and item embeddings, contradicting the goal of providing personalized recommendations. While existing research predominantly ascribes this flaw to the over-smoothing problem, this paper underscores the critical, often overlooked role of the over-correlation issue in diminishing the effectiveness of GNN representations and subsequent recommendation performance. The unclear relationship between over-correlation and over-smoothing in RS, coupled with the challenge of adaptively minimizing the impact of over-correlation while preserving collaborative filtering signals, is quite challenging. To this end, this paper aims to address the aforementioned gap by undertaking a comprehensive study of the over-correlation issue in graph collaborative filtering models. Empirical evidence substantiates the widespread prevalence of over-correlation in these models. Furthermore, a theoretical analysis establishes a pivotal connection between the over-correlation and over-smoothing predicaments. Leveraging these insights, we introduce the Adaptive Feature De-correlation Graph Collaborative Filtering (AFDGCF) Framework, which dynamically applies correlation penalties to the feature dimensions of the representation matrix, effectively alleviating both over-correlation and over-smoothing challenges. The efficacy of the proposed framework is corroborated through extensive experiments conducted with four different graph collaborative filtering models across four publicly available datasets, demonstrating the superiority of AFDGCF in enhancing the performance landscape of graph collaborative filtering models.

## 1 INTRODUCTION

Recommender systems (RS) have been widely adopted in various online applications, aiming to tackle the challenge of information overload by offering personalized item recommendations (Wu et al., 2022a). Collaborative filtering (CF) has been a popular RS technique that learns informative user and item representations from historical interactions. With the rapid development of graph neural networks (GNNs) and their success in processing graph-structured data, GNN-based CF has become prominent in RS research, capturing collaborative signals in the high-order connectivity between users and items (Wang et al., 2019). Despite the remarkable achievements facilitated by GNNs in RS, significant challenges persist in attaining high robustness and accuracy, such as embedding over-correlation and over-smoothing problems.

In the context of GNN-based recommendation models, a single GNN layer predominantly considers the immediate neighboring nodes of users and items, potentially limiting the ability to capture deep collaborative signals. Addressing this limitation, conventional GNN-based recommendation models stack multiple GNN layers to expand their receptive fields. However, this practice can lead to performance degradation as the number of stacked layers increases (Wang et al., 2019; Zhao & Akoglu, 2019). Prior studies on GNN-based CF often attribute this degradation to the widely discussed over-smoothing problem, where node representations tend to converge towards similarity with escalating layer count (Zhao & Akoglu, 2019; Liu et al., 2020; Chen et al., 2020b; Rusch et al., 2023). However, it is important to note that another critical factor contributing to performance decline is the feature over-correlation issue. As the number of GNN layers grows, the dimensions of node representations' features become increasingly correlated (Jin et al., 2022), negatively impact-

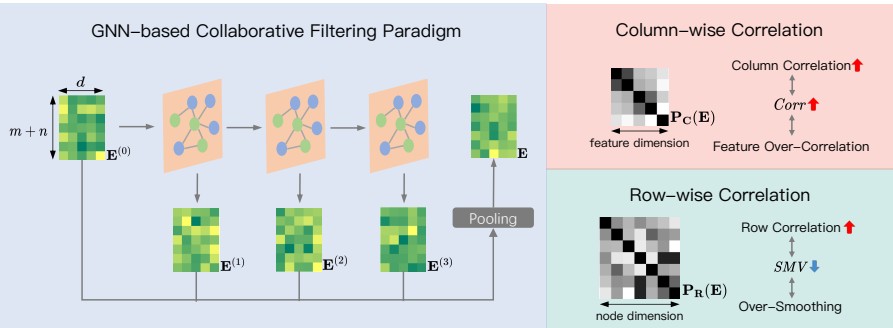

Figure 1: The illustration of the GNN-based CF, over-correlation and over-smoothing issues.

ing embedding learning quality. While over-correlation and over-smoothing share some connection (to be theoretically discussed in the Methodology Section), their primary distinction lies in their focus on relationships: over-smoothing pertains to relationships between node representations (in the row direction of the representation matrix), while over-correlation pertains to relationships between feature dimensions of the representations (in the column direction of the representation matrix).

In the literature, while various GNN-based CF studies, including GCMC (Berg et al., 2017), SpectralCF (Zheng et al., 2018), PinSage (Ying et al., 2018), and NGCF (Wang et al., 2021), have primarily concentrated on GNN structures, they have often overlooked the aforementioned issues. Subsequent investigations (He et al., 2020; Chen et al., 2020c; Mao et al., 2021; Liu et al., 2021; Peng et al., 2022; Xia et al., 2022; Cai et al., 2023; Xia et al., 2023) have started addressing these problems, with an emphasis on mitigating the over-smoothing issue but largely neglecting feature over-correlation. Consequently, several unresolved challenges remain in GNN-based recommender systems. Firstly, the influence of feature over-correlation on RS and its relationship with over-smoothing remains unclear. Secondly, devising dedicated strategies to alleviate feature over-correlation in GNN-based recommendation models remains an unexplored but essential avenue for enhancing embedding capability. Lastly, the varying severity of over-correlation and over-smoothing across different GNN layers underscores the importance of crafting layer-wise adaptive techniques to appropriately control feature learning.

To address these challenges, in this paper, we propose a comprehensive de-correlation paradigm for GNN-based recommendation models, named the Adaptive Feature De-correlation Graph Collaborative Filtering (AFDGCF) Framework. To the best of our knowledge, this is the first investigation into the impact of feature over-correlation issue in RS. We commence by providing empirical validation, demonstrating the prevalence of the feature over-correlation issue in GNN-based methods. Subsequently, through theoretical analysis, we establish a fundamental connection between feature over-correlation and over-smoothing. Building upon this insight, we propose the model-agnostic AFDGCF framework, which adaptively penalizes correlations among feature dimensions within the output representations of each GNN layer. Our approach's efficacy is extensively validated through the experiments on different implementations employing diverse GNN-based CF methods and various publicly available datasets. These results demonstrate our AFDGCF framework's general effectiveness in alleviating both the over-correlation and over-smoothing issues, and thus enhancing model performance.

## 2 RELATED WORK

This subsection presents pertinent literature relevant to the paper, encompassing GNN-based CF models and studies investigating over-correlation and over-smoothing issues.

### 2.1 GNN-BASED COLLABORATIVE FILTERING

Within the realm of RS, GNN-based CF emerges as a forefront avenue of research. It leverages the inherent higher-order connectivity present in historical user-item interactions to attain enhanced performance compared to conventional matrix factorization techniques (Mnih & Salakhutdinov, 2007; Koren et al., 2009). During the nascent phase, Li & Chen pioneered the exploration of the recommendation problem through a graph-oriented approach, treating CF as a link prediction task within bipartite graphs. Subsequently, propelled by the progression of GNNs, researchers progressively delved into GNNs' integration within the realm of RS. As an illustration, GCMC (Berg et al., 2017) established a graph-based auto-encoder framework employing GNNs to facilitate the augmentation

of the rating matrix. SpectralCF (Zheng et al., 2018) directly performed spectral domain learning on the user-item bipartite graph, alleviating the cold-start problem in CF. By leveraging efficient random walks, PinSage (Ying et al., 2018) was the first to apply GNNs to industrial-scale RS. NGCF (Wang et al., 2021) injected the collaborative signal latent in the high-order connectivity between users and items into embeddings via propagation on the user-item graph. Building on this, LightGCN (He et al., 2020) and GCCF (Chen et al., 2020c) subsequently streamlined the message-passing process, devising GNN paradigms tailored for CF tasks. Afterward, HMLET (Kong et al., 2022) attempted to amalgamate linear and non-linear methodologies by applying a gating module to choose linear or non-linear propagation for each user and item.

## 2.2 OVER-SMOOTHING AND OVER-CORRELATION

The proliferation of layers in GNNs gives rise to two significant challenges: over-smoothing and over-correlation. The former leads to highly similar node representations, making it difficult to distinguish between them, while the latter results in redundant information among feature representations. Both issues elevate the risk of over-fitting in GNN-based models and curtail their expressive capacity. Currently, in the field of GNNs, numerous studies have been dedicated to exploring how these two issues affect the performance of deeper GNNs in graph-related tasks (Zhao & Akoglu, 2019; Liu et al., 2020; Chen et al., 2020b; Rusch et al., 2023; Jin et al., 2022). In the realm of RS, various studies have also endeavored to mitigate the over-smoothing issue. One category of methodologies aims to streamline the process of message passing. For instance, LightGCN (He et al., 2020) eliminated feature transformation and non-linear activation from GNNs, and GCCF (Chen et al., 2020c) incorporated a residual network structure while discarding non-linear activation. UltraGCN (Mao et al., 2021) skipped explicit message-passing through infinite layers. SVD-GCN (Peng et al., 2022) substituted graph convolutions with truncated SVD. The second set of approaches performs feature propagation within subdivided sub-graphs. For example, IMP-GCN (Liu et al., 2021) employed graph convolutions within sub-graphs composed of similar-interest users and the items they interacted with. Similarly, Xia et al. proposed a graph resizing technique that recursively divides a graph into sub-graphs. Furthermore, additional investigations (Xia et al., 2022; Cai et al., 2023; Xia et al., 2023) endeavored to tackle the over-smoothing challenge in GNN-based CF using strategies including contrastive learning and knowledge distillation. Besides the aforementioned efforts to alleviate over-smoothing, it is noticeable that over-smoothing may not invariably be detrimental to RS. This is because the smoothness of embeddings plays a pivotal role in the effectiveness of GNN-based CF models (He et al., 2020).

Although significant progress has been made in the research of over-smoothing, there is currently a lack of studies on the over-correlation issue in GNN-based CF. In this paper, we analyze the impact of the over-correlation issue on GNN-based CF methods and propose suitable solutions.

## 3 PRELIMINARIES

In this section, we will first introduce the paradigm of GNN-based CF. Then, we present the metrics for measuring over-correlation and over-smoothing issues. Subsequently, we explore the manifestations of the two issues mentioned above in GNN-based CF models.

### 3.1 GNN-BASED COLLABORATIVE FILTERING PARADIGM

In general, the input of CF methods consists of a user set $\mathcal{U} = \{u_1, u_2, \cdots, u_m\}$, an item set $\mathcal{I} = \{i_1, i_2, \cdots, i_n\}$, and the interaction matrix between users and items $\mathbf{R} \in \{0, 1\}^{m \times n}$, where each element $r_{u,i} \in \mathbf{R}$ represents the interaction behavior of user $u$ with item $i$, such as purchase, click, etc., using 0 and 1 to indicate. In GNN-based CF, the interaction matrix above is reformulated into a user-item graph $\mathcal{G} = <\mathcal{U} \cup \mathcal{I}, \mathbf{A}>$, where $\mathbf{A}$ represents the adjacency matrix:

$$\mathbf{A} = \begin{bmatrix} \mathbf{0} & \mathbf{R} \\ \mathbf{R}^\mathrm{T} & \mathbf{0} \end{bmatrix}. \tag{1}$$

Aggregation operations and update operations can represent the message-passing process on the user-item graph (Wu et al., 2022b):

$$\begin{aligned} \mathbf{e}_i^{(l+1)} &= \mathrm{Updater}(\mathbf{e}_i^{(l)}, \mathrm{Aggregator}\left(\{\mathbf{e}_u^l, \forall u \in \mathcal{N}_i\}\right)) \\ \mathbf{e}_u^{(l+1)} &= \mathrm{Updater}(\mathbf{e}_u^{(l)}, \mathrm{Aggregator}\left(\{\mathbf{e}_i^l, \forall i \in \mathcal{N}_u\}\right)) \end{aligned}, \tag{2}$$

where $\mathbf{e}_u^{(l)}, \mathbf{e}_i^{(l)}$ represents the representation of users or items at layer $l$. $\mathcal{N}_u, \mathcal{N}_i$ represent the neighboring nodes of user $u$ or item $i$, respectively. Aggregator denotes the function responsible for

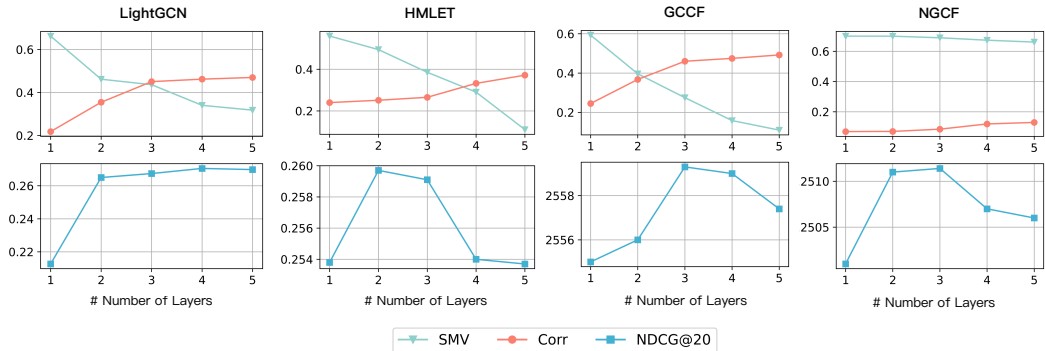

Figure 2: i) The recommendation performance (measured by NDCG@20); ii) The feature correlation of the representations (measured by *Corr*); iii) The smoothness of the representations (measured by *SMV*) learned by state-of-the-art GNN-based CF models on Movielens dataset.

aggregation operations, such as mean-pooling and attention mechanisms. Updator refers to the operation function that updates the current node representation, for instance, sum operation and nonlinear transformation.

After the message-passing process, node representation matrix:

$$\mathbf{E}^{(l)} = \left[ \mathbf{e}_{u_1}^{(l)}, \cdots, \mathbf{e}_{u_m}^{(l)}, \mathbf{e}_{i_1}^{(l)}, \cdots, \mathbf{e}_{i_n}^{(l)} \right] \tag{3}$$

can be obtained at each layer (as shown in Figure 1). The final node representation matrix $\mathbf{E}$ can be obtained through various pooling operations, such as mean/sum/weighted-pooling and concatenation:

$$\mathbf{E} = \text{Pooling}(\mathbf{E}^{(0)}, \mathbf{E}^{(1)}, \cdots, \mathbf{E}^{(L)}), \tag{4}$$

where $L$ denotes the number of layers, $\mathbf{E}^{(0)}$ represents learnable embedding matrix. Without losing generalization, this paper's analysis will focus on LightGCN (He et al., 2020), one of the most representative methods in GNN-based CF. Its message-passing process can be formulated as follows:

$$\mathbf{E}^{(l+1)} = (\mathbf{D}^{-\frac{1}{2}}\mathbf{A}\mathbf{D}^{-\frac{1}{2}})\mathbf{E}^{(l)}, \tag{5}$$

where $\mathbf{D}$ is the degree matrix of $\mathbf{A}$.

## 3.2 Over-smoothing and Over-Correlation

Over-smoothing issue refers to the growing similarity between the node representations of the GNN model's output as the number of GNN layers increases. Over-correlation issue refers to the increasing correlation between the feature dimensions of the GNN model's output representations as the number of GNN layers increases. The former will lead to convergence of node representations, making them difficult to be distinguished, while the latter will lead to feature redundancy among the learned representations. In recommendation systems, both of them will constrain the quality of learned user and item representations, increase the risk of over-fitting, and result in sub-optimal model performance. To investigate their impact on GNN-based CF, we adopt metrics proposed in previous studies within the GNN domain to assess them.

*SMV* (Liu et al., 2020) is utilized to gauge the similarity between user and item representations, which computes the average normalized Euclidean distance between any two nodes:

$$SMV(\mathbf{E}) = \frac{1}{m(m-1)} \sum_{i,j \in \mathcal{U} \cup \mathcal{I}, i \neq j} D\left(\mathbf{E}_{i*}, \mathbf{E}_{j*}\right), \tag{6}$$

where $D(x, y)$ denotes the normalized euclidean distance between vector $x$ and $y$.

*Corr* (Jin et al., 2022) is utilized to measure the correlation between feature dimensions, which calculates the average Pearson correlation coefficient (Cohen et al., 2009) between any two feature dimensions:

$$Corr(\mathbf{E}) = \frac{1}{d(d-1)} \sum_{i \neq j} |\rho(\mathbf{E}_{*i}, \mathbf{E}_{*j})| \quad i, j \in \{1, 2, \ldots, d\}, \tag{7}$$

where $d$ represents the dimension of the features, $\mathbf{E}_{*i}$ denotes the i-th feature of users and items, and $\rho(x, y)$ represents the Pearson correlation coefficient between vectors $x$ and $y$, ranging from $-1$ to $1$. The larger the absolute value, the stronger the correlation between the corresponding vectors.

Based on the two metrics above, we explore over-correlation and over-smoothing issues in GNN-based CF models. From Figure 2, we observe the following trends for these four models: (1) *Corr*

increases with the number of layers, indicating an increase in the correlation among the learned representations. (2) *SMV* decreases with the number of layers, indicating an increase in the smoothness among the learned representations. (3) The recommendation performance does not consistently improve with the increase in the number of layers; instead, performance degradation occurs after reaching 2 to 3 layers. The above three points reveal the widespread existence of over-correlation and over-smoothing issues in GNN-based CF models, leading to the suboptimal performance of the models. Moreover, the severity of both issues increases with the number of layers, indicating a certain association between them. In the next section, we will analyze their association theoretically and propose solutions.

## 4 METHODOLOGY

In this section, we begin by analyzing the theoretical connection between the over-correlation and over-smoothing issue. Building upon this analysis, we propose an Adaptive Feature Decorrelation Graph Collaborative Filtering (AFDGCF) Framework, which addresses the issues of over-correlation and over-smoothing in GNN-based CF simultaneously while finding a trade-off between over-smoothing and representation smoothness.

### 4.1 ANALYSIS ON OVER-CORRELATION AND OVER-SMOOTHING

In this subsection, we endeavor to analyze the relationship between over-correlation and over-smoothing theoretically. Firstly, we define the concepts of column correlation and row correlation of the representation matrix, which correspond to the feature correlation and smoothness, respectively. Then, through mathematical proof, we demonstrate a proportional relationship between the column correlation and row correlation of the matrix, thus inferring the connection between over-correlation and over-smoothing issues.

In general, when calculating row or column correlation coefficients, it is common practice to first normalize the matrix to have a mean of $0$ and a variance of $1$ (Cohen et al., 2009). Here, to simplify the analysis process furthermore, we assume that the representation matrix $\mathbf{E}$ satisfies double standardization, *i.e.*, each row and column of $\mathbf{E}$ have a mean of $0$ and a variance of $1$. Under such assumptions, we can define the column correlation coefficient matrix of $\mathbf{E}$ as $\mathbf{P_C} = \frac{1}{m}\mathbf{E}^T\mathbf{E}$, where $\mathbf{P_C}_{ij} = \rho(\mathbf{E}_{*i}, \mathbf{E}_{*j})$. Similarly, we can define the row correlation coefficient matrix of $\mathbf{E}$ as $\mathbf{P_R} = \frac{1}{n}\mathbf{E}\mathbf{E}^T$ where $\mathbf{P_R}_{ij} = \rho(\mathbf{E}_{i*}, \mathbf{E}_{j*})$. Based on this, we can use $\|\mathbf{P_C}\|_F$ and $\|\mathbf{P_R}\|_F$ to describe the column correlation and row correlation of matrix $\mathbf{E}$, respectively. In fact, the relationship between row correlation and column correlation of a matrix has been studied in previous research (Efron, 2008). Next, we will demonstrate the association between them based on this.

THEOREM 1. *If matrix $\mathbf{E}$ satisfies double standardization, then its corresponding row correlation matrix $\mathbf{P_C}$ and column correlation matrix $\mathbf{P_R}$ satisfy $\|\mathbf{P_R}\|_F \propto \|\mathbf{P_C}\|_F$.*

*Proof.* The singular value decomposition of matrix $\mathbf{E}$ is:
$$\mathbf{E}_{m \times n} = \mathbf{U}_{m \times d} \ \mathbf{\Sigma}_{d \times d} \ \mathbf{V}^T_{d \times n}, \tag{8}$$
where $\mathbf{\Sigma}$ is the diagonal matrix of ordered singular values, $\mathbf{U}$ and $\mathbf{V}$ are orthogonal matrices satisfying $\mathbf{U}^T\mathbf{U} = \mathbf{V}^T\mathbf{V} = \mathbf{I}_d$, $\mathbf{I}_d$ is the $d \times d$ identity. The squares of the diagonal elements:
$$e_1 \geq e_2 \geq \cdots \geq e_d > 0 \quad (e_i = \mathbf{\Sigma}_i^2), \tag{9}$$
are the eigenvalues of $\mathbf{E}^T\mathbf{E} = \mathbf{V}^T\mathbf{\Sigma}^2\mathbf{V}$. Then we have:
$$\begin{aligned} \frac{\sum_{i=1}^n \sum_{j=1}^n \mathbf{P_C}_{ij}^2}{n^2} &= \frac{\text{tr}((\mathbf{E}^T\mathbf{E})^2)}{m^2n^2} = \frac{\text{tr}(\mathbf{V}^T\mathbf{\Sigma}^4\mathbf{V})}{m^2n^2} = c, \\ \frac{\sum_{i=1}^n \sum_{j=1}^n \mathbf{P_R}_{ij}^2}{m^2} &= \frac{\text{tr}((\mathbf{E}\mathbf{E}^T)^2)}{m^2n^2} = \frac{\text{tr}(\mathbf{U}\mathbf{\Sigma}^4\mathbf{U}^T)}{m^2n^2} = c, \end{aligned} \tag{10}$$

where $c = \sum_{i=1}^d e_i^2/(mn)^2$. That is to say: $\frac{1}{m}\|\mathbf{P_R}\|_F = \frac{1}{n}\|\mathbf{P_C}\|_F$. $\qquad \square$

The above theory reveals a proportional relationship between column correlation and row correlation of a matrix. In the GNN-based CF model, for the representation matrix $\mathbf{E}$, the column correlation represents the correlation between features, and the row correlation describes the similarity between user and item representations, which can be considered as a proxy for measuring smoothness. The larger the row correlation, the smaller the average difference between representations, indicating a higher level of smoothing. Therefore, we can conclude that over-correlation and over-smoothing positively correlate in degree. We can mitigate the impact on the model by controlling one of them, thus alleviating the effect of the other.

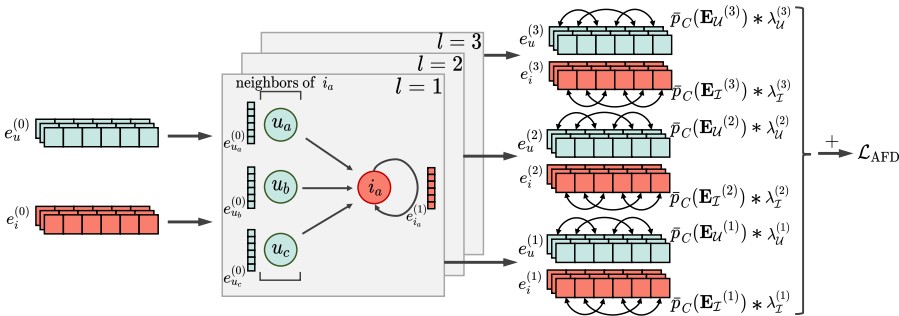

Figure 3: An overview of the AFDGCF Framework.

## 4.2 FEATURE DE-CORRELATION

To alleviate the excessively high correlation in feature dimensions of user and item representations learned by GNN-based CF models, a straightforward approach is to design a loss that penalizes the correlation between features. Our objective is to reduce the average correlation between columns of the representation matrix $\mathbf{E}$. Formally,

$$\bar{p}_C(\mathbf{E}) = \frac{1}{d(d-1)} \sum_{i \neq j} \rho(\mathbf{E}_{*i}, \mathbf{E}_{*j})^2 \quad i, j \in \{1, 2, \dots, d\}. \tag{11}$$

Since the correlation coefficient of a vector with itself is always 1, we can also represent it using the previously defined column correlation coefficient matrix $\mathbf{P_R}$:

$$\bar{p}_C(\mathbf{E}) = \sqrt{\frac{1}{2} \|\mathbf{P_R} - \mathbf{I}_d\|_F^2} = \frac{1}{\sqrt{2}} \|\mathbf{P_R} - \mathbf{I}_d\|, \tag{12}$$

where $\mathbf{I}_d$ is the $d \times d$ identity and $\mathbf{P_R}$ can be calculated using the following formula:

$$\mathbf{P_R}_{ij} = \frac{\mathbf{Cov_R}_{ij}}{\sqrt{\mathbf{Cov_R}_{ii} * \mathbf{Cov_R}_{jj}}}, \tag{13}$$

in which $\mathbf{Cov_R}$ denotes the covariance matrix among the column vectors of the representation matrix $\mathbf{E}$:

$$\mathbf{Cov_R} = (\mathbf{E} - \bar{\mathbf{E}})^T (\mathbf{E} - \bar{\mathbf{E}}), \tag{14}$$

where $\bar{\mathbf{E}}$ represents a matrix composed of the mean values of each column of $\bar{\mathbf{E}}$. It's worth noting that here we utilize the entire representation matrix to compute the correlations between feature dimensions. In fact, to reduce the computational complexity, when dealing with large-scale user and item sets, it is possible to randomly sample a batch of users and items for calculating the correlations between feature dimensions.

## 4.3 ADAPTIVE FEATURE DE-CORRELATION

Directly applying the function $\bar{p}_C(\cdot)$ to the learned representation matrix $\mathbf{E}$ of the GNN-based CF model is not the optimal choice. Firstly, we consider that different GNN-based CF models may use different pooling functions. For example, GCCF uses concatenation, while LightGCN uses mean pooling. This results in significant differences in the correlation of the representation matrix $\mathbf{E}$ for different models. For the sake of model-agnostic objectives, we propose penalizing the representations obtained after each message-passing operation as an alternative approach. Furthermore, since user and item feature distributions often exhibit substantial differences, applying penalties separately to the user and item representation matrix is necessary. This approach also helps in preserving the relationships between users and items learned by the model, thus avoiding potential disruptions. In this case, our feature de-correlation loss can be formulated in the following manner:

$$\mathcal{L}_{\text{AFD}} = \sum_{l=1}^{L} \lambda_{\mathcal{U}}^{(l)} \bar{p}_C(\mathbf{E}_{\mathcal{U}}^{(l)}) + \lambda_{\mathcal{I}}^{(l)} \bar{p}_C(\mathbf{E}_{\mathcal{I}}^{(l)}). \tag{15}$$

It is worth noting that we do not impose a penalty on $\mathbf{E}_{\mathcal{U}}^{(0)}, \mathbf{E}_{\mathcal{I}}^{(0)}$ because the 0-th layer represents the initial user and item embeddings without undergoing graph convolution, usually subject to $L_2$ regularization constraints. $\lambda_{\mathcal{U}}^{(l)}, \lambda_{\mathcal{I}}^{(l)}$ represents the penalty coefficient for the $l$-th layer corresponding to users and items. If fixed at $1/L$, it indicates a fixed penalty of the same magnitude for each layer. However, this is not the most reasonable approach. The penalty sizes accepted between different layers should be dynamically adjusted during training based on the relative correlation magnitudes.

For a GNN-based CF model with a specific number of layers, the deeper representations often have greater embedding smoothness while having higher feature correlations. Considering the contribution of feature smoothness to the effectiveness of GNN-based CF models, it is crucial to maintain the smoothness of deep representations while restricting the feature correlations of the model's representations. Therefore, we propose the following strategies:

$$\lambda_{\mathcal{U}}^{(l)} = \frac{1/\bar{p}_C(\mathbf{E}_{\mathcal{U}}^{(l)})}{\sum_{i=1}^{L} 1/\bar{p}_C(\mathbf{E}_{\mathcal{U}}^{(i)})}, \; \lambda_{\mathcal{I}}^{(l)} = \frac{1/\bar{p}_C(\mathbf{E}_{\mathcal{I}}^{(l)})}{\sum_{i=1}^{L} 1/\bar{p}_C(\mathbf{E}_{\mathcal{I}}^{(i)})}, \tag{16}$$

which allocates lower penalty coefficients to deeper representations and maintains the total penalty amount. If the adaptive strategy is not employed, throughout the entire training process, the penalty coefficients for user and item correspondence at each layer remain fixed and unchanging. However, with the adoption of the adaptive strategy, the penalty coefficients for user and item correspondence at each layer will dynamically change in each training step and tend to stabilize as the model approaches convergence. The final loss function is shown below:

$$\mathcal{L} = \mathcal{L}_{\text{CF}} + \alpha \mathcal{L}_{\text{AFD}}, \tag{17}$$

where $\mathcal{L}_{\text{CF}}$ represents the original loss functions of the CF model like Bayesian Personalized Ranking (BPR) loss (Rendle et al., 2012), Cross-Entropy (CE) loss, etc. $\alpha$ is a hyper-parameter used to adjust the relative magnitude of the adaptive feature de-correlation loss.

## 5 EXPERIMENTS

To better understand the capabilities and effectiveness of our proposed AFDGCF framework, we implement it using four representative GNN-based CF models and evaluate its performance on four publicly available datasets. [1] Specifically, we will answer the following research questions to unfold the experiments: 1) **RQ1**: What degree of performance improvement can our proposed AFDGCF framework bring to the state-of-the-art methods in GNN-based CF? 2) **RQ2**: Does adaptive de-correlation have a performance advantage compared to fixed one? 3) **RQ3**: How do the hyper-parameters influence the effectiveness of the proposed AFDGCF framework? and; 4) **RQ4**: What impact does the AFDGCF framework have on the correlation of the learned representations?

### 5.1 EXPERIMENTAL SETTINGS

**Datasets and metrics.** We selected four publicly available datasets to evaluate the performance of the proposed AFDGCF, including MovieLens (Harper & Konstan, 2015), Yelp [2], Gowalla (Cho et al., 2011), and Amazon-book (McAuley et al., 2015), which vary in domain, scale, and density. For the Yelp and Amazon-book datasets, we excluded users and items with fewer than 15 interactions to ensure data quality. Similarly, for the Gowalla dataset, we removed those with fewer than 10 interactions. Moreover, we partitioned each dataset into training, validation, and testing sets using an 8:1:1 ratio. We chose Recall@K, Normalized Discounted Cumulative Gain (NDCG)@K, and Mean Average Precision (MAP)@K as the evaluation metrics (K=10). We also employed the all-ranking protocol (He et al., 2020) to avoid bias arising from sampling.

**Methods for Comparison.** To verify the effectiveness of our proposed method, we compared it with several state-of-the-art models, including BPRMF (Rendle et al., 2012), DMF (Xue et al., 2017), ENMF (Chen et al., 2020a), MultiVAE (Liang et al., 2018), RecVAE (Shenbin et al., 2020), NGCF (Wang et al., 2019), and GCCF (Chen et al., 2020c).

**Implementation Details.** For a fair comparison, we employed the Adam optimizer for all models with a learning rate of $1e^{-3}$ and set the training batch size to 4096. All models have an embedding size or hidden dimension set to 128, and model parameters were initialized using the Xavier distribution. Furthermore, we used early stopping for the indicator NDCG@10 to prevent the model from over-fitting. For the four GNN-based CF models, we performed 3-layer propagation. As for the two VAE-based models, we set their encoder and decoder as a 1-hidden-layer MLP with $[n, 600, 128, 600, n]$ dimensions. Hyper-parameters for all baseline models were carefully tuned. Specifically, the weight of L2 regularization is optimized within the range $\{1e^{-2}, 1e^{-3}, 1e^{-4}, 1e^{-5}, 1e^{-6}\}$, and dropout ratios were selected from $\{0.0, 0.1, \cdots, 0.8, 0.9\}$. Other model-specific hyper-parameters were fine-tuned within their recommended ranges. For the proposed AFDGCF framework, we tuned the hyper-parameter $\alpha$ within the range $\{1e^{-5}, 5e^{-5}, \cdots, 5e^{-2}, 1e^{-1}\}$. The implementation of all models and their evaluation was conducted using the open-source framework framework Recbole (Zhao et al., 2021).

---

[1] All the code will be publicly available after the paper is accpeted.

[2] https://www.yelp.com/dataset

| Dataset | MovieLens | | | Yelp | | | Gowalla | | | Amazon-book | | |
|---|---|---|---|---|---|---|---|---|---|---|---|---|
| Method | Recall | NDCG | MAP | Recall | NDCG | MAP | Recall | NDCG | MAP | Recall | NDCG | MAP |
| BPRMF | 0.1697 | 0.2353 | 0.1347 | 0.0643 | 0.0437 | 0.0240 | 0.1110 | 0.0790 | 0.0509 | 0.0678 | 0.0470 | 0.0273 |
| DMF | 0.1739 | 0.2338 | 0.1320 | 0.0560 | 0.0375 | 0.0207 | 0.0828 | 0.0589 | 0.0371 | 0.0643 | 0.0447 | 0.0260 |
| ENMF | 0.1941 | 0.2613 | 0.1524 | 0.0742 | 0.0520 | 0.0298 | 0.1271 | 0.0914 | 0.0597 | 0.0839 | 0.0617 | 0.0374 |
| MultiVAE | 0.1889 | 0.2370 | 0.1318 | 0.0731 | 0.0494 | 0.0275 | 0.1188 | 0.0838 | 0.0550 | 0.0816 | 0.0574 | 0.0352 |
| RecVAE | 0.1852 | 0.2524 | 0.1475 | 0.0777 | 0.0536 | 0.0301 | 0.1466 | 0.1046 | 0.0695 | 0.1044 | 0.0772 | 0.0484 |
| NGCF | 0.1743 | 0.2404 | 0.1376 | 0.0613 | 0.0419 | 0.0229 | 0.1129 | 0.0800 | 0.0519 | 0.0695 | 0.0484 | 0.0287 |
| **AFD-NGCF** | **0.1764** | **0.2449** | **0.1416** | **0.0636** | **0.0432** | **0.0237** | **0.1233** | **0.0869** | **0.0566** | **0.0735** | **0.0511** | **0.0304** |
| **Improv.** | **1.20%** | **1.87%** | **2.91%** | **3.75%** | **3.10%** | **3.49%** | **9.21%** | **8.63%** | **9.05%** | **5.76%** | **5.58%** | **5.92%** |
| GCCF | 0.1761 | 0.2460 | 0.1431 | 0.0658 | 0.0451 | 0.0247 | 0.1268 | 0.0907 | 0.0589 | 0.0898 | 0.0639 | 0.0387 |
| **AFD-GCCF** | **0.1882** | **0.2566** | **0.1505** | **0.0739** | **0.0506** | **0.0284** | **0.1315** | **0.0949** | **0.0621** | **0.0977** | **0.0698** | **0.0427** |
| **Improv.** | **6.87%** | **4.31%** | **5.17%** | **12.31%** | **12.20%** | **14.98%** | **3.71%** | **4.63%** | **5.43%** | **8.80%** | **9.23%** | **10.34%** |
| HMLET | 0.1799 | 0.2479 | 0.1433 | 0.0723 | 0.0503 | 0.0285 | 0.1427 | 0.1028 | 0.0679 | 0.0967 | 0.0699 | 0.0430 |
| **AFD-HMLET** | **0.1922** | **0.2606** | **0.1523** | **0.0810** | **0.0562** | **0.0319** | **0.1541** | **0.1101** | **0.0727** | **0.1039** | **0.0760** | **0.0471** |
| **Improv.** | **6.84%** | **5.12%** | **6.28%** | **12.03%** | **11.73%** | **11.93%** | **7.99%** | **7.10%** | **7.07%** | **7.45%** | **8.73%** | **9.53%** |
| LightGCN | 0.1886 | 0.2540 | 0.1470 | 0.0756 | 0.0525 | 0.0295 | 0.1433 | 0.1019 | 0.0664 | 0.1015 | 0.0744 | 0.0465 |
| **AFD-LightGCN** | **0.1985** | **0.2689** | **0.1594** | **0.0831** | **0.0575** | **0.0327** | **0.1564** | **0.1117** | **0.0736** | **0.1078** | **0.0781** | **0.0486** |
| **Improv.** | **5.25%** | **5.87%** | **8.44%** | **9.92%** | **9.52%** | **10.85%** | **9.14%** | **9.62%** | **10.84%** | **6.21%** | **4.97%** | **4.52%** |

Table 1: Performance comparisons on four datasets. (Underlines represent the optimal performance)

| Method | Metric | MovieLens | Yelp | Gowalla | Amazon-book |
|---|---|---|---|---|---|
| HMLET | Epochs | 167 | 189 | 347 | 458 |
| | Time/Epoch | 5.95s | 9.54s | 10.25s | 58.12s |
| AFD-HMLET | Epochs | 112 | 109 | 205 | 198 |
| | Time/Epoch | 6.87s | 10.58s | 11.27s | 61.23s |
| LightGCN | Epochs | 309 | 364 | 396 | 742 |
| | Time/Epoch | 4.51s | 6.76s | 6.98s | 32.06s |
| AFD-LightGCN | Epochs | 114 | 151 | 302 | 360 |
| | Time/Epoch | 5.12s | 7.31s | 7.82s | 35.73s |

Table 2: Comparison of training efficiency.

| Dataset | Metric | AFD-NGCF | AFD-f-NGCF | AFD-GCCF | AFD-f-GCCF |
|---|---|---|---|---|---|
| Yelp | Recall | **0.0636** | 0.0623 | **0.0739** | 0.0719 |
| | NDCG | **0.0432** | 0.0424 | **0.0506** | 0.0487 |
| Gowalla | Recall | **0.1233** | 0.1198 | **0.1315** | 0.1298 |
| | NDCG | **0.0869** | 0.0850 | **0.0949** | 0.0937 |
| Dataset | Metric | AFD-HMLET | AFD-f-HMLET | AFD-LightGCN | AFD-f-LightGCN |
| Yelp | Recall | **0.0810** | 0.0796 | **0.0831** | 0.0808 |
| | NDCG | **0.0562** | 0.0552 | **0.0575** | 0.0558 |
| Gowalla | Recall | **0.1541** | 0.1465 | **0.1564** | 0.1505 |
| | NDCG | **0.1101** | 0.1069 | **0.1117** | 0.1081 |

Table 3: Ablation study of AFDGCF.

## 5.2 PERFORMANCE COMPARISONS (RQ1)

Table 1 demonstrates the effectiveness of our proposed AFDGCF framework as instantiated on four GNN-based CF models and provides a comparative analysis against various baseline methods across four distinct datasets.The table reveals that implementing AFDGCF in all the aforementioned GNN-based CF models is effective. This stems from notable performance enhancements achieved by mitigating the over-correlation issue, while successfully balancing mitigating over-smoothing and maintaining embedding smoothness. Furthermore, the AFDGCF framework demonstrates even more pronounced performance enhancements on large-scale datasets. On Yelp, for Recall@10, AD-FGCF achieves performance improvements of 3.75%, 12.31%, 12.03%, and 9.92% over NGCF, GCCF, HMLET, and LightGCN, respectively. It is worth noting that in our experimental setup, we standardized the embedding dimension for all GNN-based CF methods to 128 and conducted three rounds of feature propagation. As a result, HMLET, contrary to its original design, does not outperform LightGCN on most datasets. Moreover, we can observe that most GNN-based CF models trail slightly behind VAE-based models in performance. This indicates that GNN-based CF models might be prone to overfitting when dealing with extensive sparse interactions, a phenomenon linked to the two issues highlighted in this paper.

Recent studies have indicated that GNN-based CF models, like LightGCN, are much more difficult to train than traditional MF models (Peng et al., 2022; Wang et al., 2022). These models often require a larger number of epochs to reach the optimal state. We hypothesize that issues of over-correlation and over-smoothing diminish the distinction between positive and negative samples, thereby impacting learning efficiency. Fortunately, we found our AFDGCF framework can alleviate this phenomenon. As evidenced in Table 2, within the AFDGCF framework, both LightGCN and HMLET attain optimal performance with fewer epochs on various datasets. In some cases, the required epochs are reduced by half.

## 5.3 ABLATION STUDY OF AFDGCF (RQ2)

A key component of our proposed AFDGCF is the adaptive strategy. Details about the variation of penalty coefficients throughout the training process can be found in the appendix. To validate this strategy, we conducted an ablation study to analyze its influence on performance. The experimental results are presented in Table 3, where AFD-f denotes applying a fixed penalty coefficient of $1/l$ to each layer within the AFDGCF framework. Clearly, we can observe that our adaptive approach consistently outperforms the fixed penalty coefficient across all GNN-based models, demonstrating the effectiveness of applying varying de-correlation penalties to distinct layers.

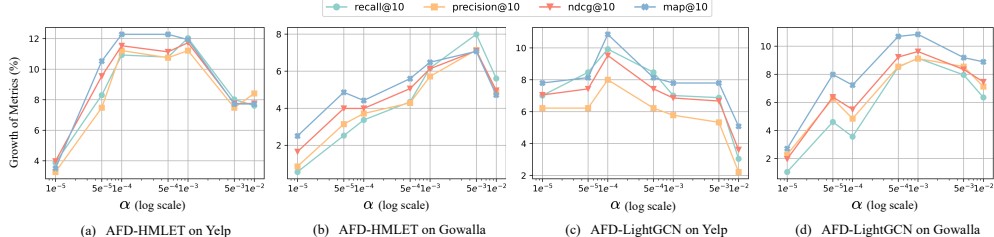

Figure 4: Performance comparison *w.r.t.* different $\alpha$. Using the results of the original model (*i.e.*, $\alpha = 0$) as the reference.

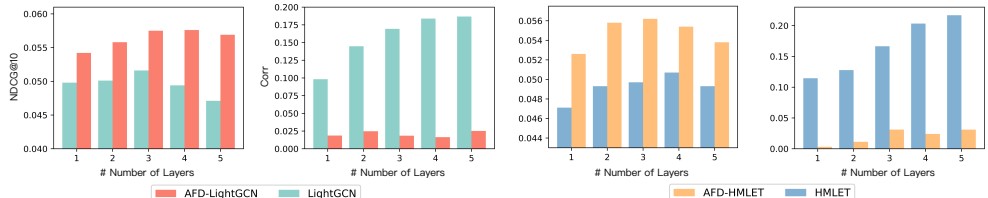

Figure 5: Comparison of GNN-based CF models with different layers before and after applying the AFDGCF framework in terms of NDCG@10, *Corr* on Yelp Dataset.

### 5.4 HYPER-PARAMETER ANALYSIS (RQ3)

In our AFDGCF framework, hyper-parameter $\alpha$ balances the CF task loss with the proposed adaptive feature de-correlation loss, acting as a regulator for the intensity of de-correlation. Figure 4 reports the performance of AFDGCF as $\alpha$ varies across the range $\{1e^{-5}, 5e^{-5}, \cdots, 5e^{-2}, 1e^{-1}\}$. We can observe that setting an appropriate $\alpha$ can markedly enhance AFDGCF's performance. Notably, the ideal $\alpha$ varies between datasets, but remains fairly consistent for different models within the same dataset. This is attributed to our design of applying de-correlation penalties to each layer rather than the final representation.

### 5.5 CASE STUDY (RQ4)

We qualitatively investigate the effects of our proposed AFDGCF framework using LightGCN and HMLET as examples in the following two aspects: i) the de-correlation effect on GNN-based CF models with different layers, and ii) the resulting performance enhancements.

In Figure 5, We have reported a comparison of the performance and the feature correlations of Light-GCN and HMLET (with layers ranging from 1 to 5) before and after applying the AFD penalty. Observations can be made as follows: 1) For any number of layers, the AFDGCF framework improves performance, and larger layers correspond to higher performance gains. 2) Both LightGCN and AFD-LightGCN achieve optimal performance at 3 layers, with performance degradation at 4 and 5 layers. AFD-LightGCN mitigates the performance degradation significantly compared to Light-GCN, demonstrating that our proposed AFDGCF framework can alleviate the performance decline caused by over-correlation. 3) The *Corr* metric of the representations learned by AFD-LightGCN and AFD-HMLET from layers 1 to 5 is much lower than that of LightGCN and HMLET, indicating that AFDGCF effectively achieves its goal of controlling the correlation between feature dimensions in the representation matrix.

## 6 CONCLUSION

This research undertook an in-depth exploration of the largely neglected issue of over-correlation in GNN-based CF models, substantiating its widespread presence and associated degradation in model performance. Our investigations established a direct and positive correlation between over-correlation and over-smoothing, highlighting their combined detrimental impact on the effectiveness of GNN-based CF models. Specifically, we introduced the AFDGCF framework, a novel approach designed to mitigate the influence of over-correlation by strategically managing feature correlations. This was achieved through a hierarchical adaptive strategy, ensuring an optimal balance between minimizing over-smoothing and maintaining essential representation smoothness, a critical aspect for the success of GNN-based CF models. Empirical validations underscored the efficacy of the AFDGCF framework, demonstrating notable performance enhancements across different layers of various GNN-based CF models.

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

## A STATISTICAL INFORMATION OF DATASETS

The statistical information for these four datasets is summarized in Table A1. These datasets differ in domain, scale, and density.

| Datasets | User # | Item # | Interaction # | Density |
|---|---|---|---|---|
| Movielens | 6,040 | 3,629 | 836,478 | $3.8e^{-2}$ |
| Yelp | 26,752 | 19,857 | 1,001,901 | $1.8e^{-3}$ |
| Gowalla | 29,859 | 40,989 | 1,027,464 | $8.4e^{-4}$ |
| Amazon-book | 58,145 | 58,052 | 2,517,437 | $7.5e^{-4}$ |

Table A1: Statistics of the utilized datasets.

## B ADDITIONAL EXPERIMENT RESULT

### B.1 ADDITIONAL RESULT OF PERFORMANCE COMPARISON EXPERIMENT

Table B1,B2,B3 and B4 respectively present a comprehensive performance comparison of various models on the Movielens, Yelp, Gowalla, and Amazon-book datasets. It can be observed that, across all four datasets, our proposed AFDGCF framework consistently achieves significant performance improvements for GNN-based CF models by alleviating the over-smoothing issue in terms of Recall@K, Precision@K, NDCG@K, and MAP@K (K=10, 20, 50) metrics.

| | Recall@K | | | Precision@K | | | NDCG@K | | | MAP@K | | |
|---|---|---|---|---|---|---|---|---|---|---|---|---|
| | K=10 | K=20 | K=50 | K=10 | K=20 | K=50 | K=10 | K=20 | K=50 | K=10 | K=20 | K=50 |
| BPRMF | 0.1697 | 0.2591 | 0.4212 | 0.1769 | 0.1403 | 0.0956 | 0.2353 | 0.2463 | 0.2915 | 0.1347 | 0.1224 | 0.1270 |
| DMF | 0.1739 | 0.2667 | 0.4300 | 0.1748 | 0.1396 | 0.0960 | 0.2338 | 0.2477 | 0.2944 | 0.132 | 0.1223 | 0.1285 |
| ENMF | 0.1941 | 0.2888 | 0.4499 | 0.1903 | 0.1492 | 0.0996 | 0.2613 | 0.2731 | 0.3182 | 0.1524 | 0.1405 | 0.1460 |
| MultiVAE | 0.1889 | 0.2802 | 0.4443 | 0.1739 | 0.1396 | 0.0958 | 0.237 | 0.2538 | 0.3026 | 0.1318 | 0.1257 | 0.1338 |
| RecVAE | 0.1852 | 0.2742 | 0.4307 | 0.1854 | 0.1449 | 0.0964 | 0.2524 | 0.262 | 0.3043 | 0.1475 | 0.1338 | 0.1374 |
| NGCF | 0.1743 | 0.2632 | 0.4264 | 0.1814 | 0.1425 | 0.0973 | 0.2415 | 0.2511 | 0.2966 | 0.1392 | 0.1256 | 0.1304 |
| AFD-NGCF | **0.1764** | **0.2686** | **0.4309** | **0.1831** | **0.1446** | **0.0978** | **0.2449** | **0.2558** | **0.3009** | **0.1416** | **0.1286** | **0.1333** |
| GCCF | 0.1761 | 0.2671 | 0.4298 | 0.1828 | 0.1440 | 0.0976 | 0.2460 | 0.2559 | 0.3006 | 0.1431 | 0.1290 | 0.1333 |
| AFD-GCCF | **0.1882** | **0.282** | **0.4426** | **0.1892** | **0.1481** | **0.0989** | **0.2566** | **0.2670** | **0.3110** | **0.1505** | **0.1368** | **0.1409** |
| HMLET | 0.1799 | 0.2741 | 0.4400 | 0.1849 | 0.1453 | 0.0985 | 0.2479 | 0.2591 | 0.3055 | 0.1433 | 0.1305 | 0.1358 |
| AFD-HMLET | **0.1922** | **0.2863** | **0.4513** | **0.1918** | **0.1499** | **0.1004** | **0.2606** | **0.2714** | **0.3172** | **0.1523** | **0.1393** | **0.1446** |
| LightGCN | 0.1886 | 0.2861 | 0.4538 | 0.1873 | 0.1486 | 0.1001 | 0.2540 | 0.2674 | 0.3147 | 0.1470 | 0.1356 | 0.1417 |
| AFD-LightGCN | **0.1985** | **0.2932** | **0.4629** | **0.1969** | **0.1535** | **0.1029** | **0.2689** | **0.2790** | **0.3260** | **0.1594** | **0.1453** | **0.1507** |

Table B1: Performance comparisons on Movielens

### B.2 ADDITIONAL RESULT OF ABLATION STUDY

Adaptive strategy is an important component of our proposed AFDGCF framework, and figure B1 illustrates the penalty coefficients for each layer of user and item in the adaptive strategy. It can be observed that both users and items have significant differences in their initial penalty coefficients across layers. However, as the adaptive de-correlation penalty is applied, the penalty coefficients between layers become closer.

|  | Recall@K | | | Precision@K | | | NDCG@K | | | MAP@K | | |
|---|---|---|---|---|---|---|---|---|---|---|---|---|
|  | K=10 | K=20 | K=50 | K=10 | K=20 | K=50 | K=10 | K=20 | K=50 | K=10 | K=20 | K=50 |
| BPRMF | 0.0643 | 0.1043 | 0.1902 | 0.0190 | 0.0156 | 0.0115 | 0.0437 | 0.0563 | 0.0786 | 0.0240 | 0.0267 | 0.0301 |
| DMF | 0.0560 | 0.0910 | 0.1696 | 0.0160 | 0.0133 | 0.0100 | 0.0375 | 0.0484 | 0.0685 | 0.0207 | 0.0230 | 0.0259 |
| ENMF | 0.0742 | 0.1203 | 0.2149 | 0.0218 | 0.0177 | 0.0129 | 0.0520 | 0.0663 | 0.0908 | 0.0298 | 0.0329 | 0.0367 |
| MultiVAE | 0.0731 | 0.1162 | 0.2105 | 0.0212 | 0.0172 | 0.0126 | 0.0494 | 0.0630 | 0.0874 | 0.0275 | 0.0306 | 0.0344 |
| RecVAE | 0.0777 | 0.1257 | 0.2220 | 0.0230 | 0.0188 | 0.0134 | 0.0536 | 0.0685 | 0.0933 | 0.0301 | 0.0333 | 0.0372 |
| NGCF | 0.0613 | 0.1015 | 0.1880 | 0.0184 | 0.0153 | 0.0114 | 0.0419 | 0.0544 | 0.0768 | 0.0229 | 0.0256 | 0.0289 |
| AFD-NGCF | **0.0636** | **0.1039** | **0.1896** | **0.0189** | **0.0155** | **0.0114** | **0.0432** | **0.0557** | **0.0779** | **0.0237** | **0.0264** | **0.0298** |
| GCCF | 0.0658 | 0.1069 | 0.1930 | 0.0198 | 0.0162 | 0.0118 | 0.0451 | 0.0579 | 0.0802 | 0.0247 | 0.0275 | 0.0309 |
| AFD-GCCF | **0.0739** | **0.1174** | **0.2091** | **0.0218** | **0.0176** | **0.0128** | **0.0506** | **0.0642** | **0.0880** | **0.0284** | **0.0313** | **0.0350** |
| HMLET | 0.0723 | 0.1155 | 0.2068 | 0.0214 | 0.0174 | 0.0126 | 0.0503 | 0.0638 | 0.0874 | 0.0285 | 0.0314 | 0.0350 |
| AFD-HMLET | **0.0810** | **0.1285** | **0.2292** | **0.0238** | **0.0192** | **0.0138** | **0.0562** | **0.0709** | **0.0968** | **0.0319** | **0.0351** | **0.0391** |
| LightGCN | 0.0756 | 0.1207 | 0.2159 | 0.0225 | 0.0182 | 0.0131 | 0.0525 | 0.0666 | 0.0911 | 0.0295 | 0.0326 | 0.0364 |
| AFD-LightGCN | **0.0831** | **0.1295** | **0.2322** | **0.0243** | **0.0193** | **0.0140** | **0.0575** | **0.0720** | **0.0985** | **0.0327** | **0.0359** | **0.0401** |

Table B2: Performance comparisons on Yelp

|  | Recall@K | | | Precision@K | | | NDCG@K | | | MAP@K | | |
|---|---|---|---|---|---|---|---|---|---|---|---|---|
|  | K=10 | K=20 | K=30 | K=10 | K=20 | K=30 | K=10 | K=20 | K=30 | K=10 | K=20 | K=30 |
| BPRMF | 0.1110 | 0.1618 | 0.2608 | 0.0272 | 0.0202 | 0.0134 | 0.0790 | 0.0936 | 0.1177 | 0.0509 | 0.0542 | 0.0580 |
| DMF | 0.0828 | 0.1236 | 0.2038 | 0.0210 | 0.0158 | 0.0107 | 0.0589 | 0.0706 | 0.0903 | 0.0371 | 0.0396 | 0.0426 |
| ENMF | 0.1271 | 0.1839 | 0.2902 | 0.0315 | 0.0232 | 0.0149 | 0.0914 | 0.1078 | 0.1337 | 0.0597 | 0.0634 | 0.0676 |
| MultiVAE | 0.1303 | 0.1879 | 0.2914 | 0.0309 | 0.0228 | 0.0146 | 0.0925 | 0.1091 | 0.1343 | 0.0612 | 0.0650 | 0.0692 |
| RecVAE | 0.1466 | 0.2074 | 0.3167 | 0.0354 | 0.0256 | 0.0162 | 0.1046 | 0.1221 | 0.1489 | 0.0695 | 0.0735 | 0.0780 |
| NGCF | 0.1129 | 0.1634 | 0.2640 | 0.0273 | 0.0203 | 0.0135 | 0.0800 | 0.0947 | 0.1192 | 0.0519 | 0.0552 | 0.0591 |
| AFD-NGCF | **0.1233** | **0.178** | **0.2839** | **0.0294** | **0.0218** | **0.0143** | **0.0869** | **0.1028** | **0.1286** | **0.0566** | **0.0603** | **0.0645** |
| GCCF | 0.1268 | 0.1857 | 0.2923 | 0.0314 | 0.0235 | 0.0152 | 0.0907 | 0.1077 | 0.1337 | 0.0589 | 0.0628 | 0.0671 |
| AFD-GCCF | **0.1315** | **0.1893** | **0.3004** | **0.0328** | **0.0239** | **0.0155** | **0.0949** | **0.1114** | **0.1384** | **0.0621** | **0.0659** | **0.0704** |
| HMLET | 0.1427 | 0.2030 | 0.3116 | 0.0350 | 0.0254 | 0.0160 | 0.1028 | 0.1201 | 0.1466 | 0.0679 | 0.0718 | 0.0763 |
| AFD-HMLET | **0.1541** | **0.2180** | **0.3335** | **0.0375** | **0.0272** | **0.0172** | **0.1101** | **0.1284** | **0.1568** | **0.0727** | **0.0769** | **0.0818** |
| LightGCN | 0.1433 | 0.2066 | 0.3187 | 0.0351 | 0.0259 | 0.0165 | 0.1019 | 0.1200 | 0.1475 | 0.0664 | 0.0706 | 0.0752 |
| AFD-LightGCN | **0.1564** | **0.2234** | **0.3409** | **0.0383** | **0.0280** | **0.0177** | **0.1117** | **0.1310** | **0.1598** | **0.0736** | **0.0781** | **0.0830** |

Table B3: Performance comparisons on Gowalla

|  | Recall@K | | | Precision@K | | | NDCG@K | | | MAP@K | | |
|---|---|---|---|---|---|---|---|---|---|---|---|---|
|  | K=10 | K=20 | K=50 | K=10 | K=20 | K=50 | K=10 | K=20 | K=50 | K=10 | K=20 | K=50 |
| BPRMF | 0.0678 | 0.1064 | 0.1842 | 0.0198 | 0.0160 | 0.0116 | 0.0470 | 0.0588 | 0.0790 | 0.0273 | 0.0298 | 0.0328 |
| DMF | 0.0643 | 0.0995 | 0.1716 | 0.0186 | 0.0150 | 0.0107 | 0.0447 | 0.0555 | 0.0742 | 0.0261 | 0.0284 | 0.0312 |
| ENMF | 0.0839 | 0.1240 | 0.1998 | 0.0250 | 0.0192 | 0.0130 | 0.0617 | 0.0738 | 0.0937 | 0.0374 | 0.0399 | 0.0430 |
| MultiVAE | 0.0816 | 0.1209 | 0.1961 | 0.0223 | 0.0173 | 0.0119 | 0.0574 | 0.0695 | 0.0891 | 0.0352 | 0.0379 | 0.0409 |
| RecVAE | 0.1044 | 0.1505 | 0.2321 | 0.0290 | 0.0220 | 0.0145 | 0.0772 | 0.0893 | 0.1109 | 0.0484 | 0.0499 | 0.0534 |
| NGCF | 0.0695 | 0.1063 | 0.1777 | 0.0196 | 0.0155 | 0.0109 | 0.0484 | 0.0597 | 0.0783 | 0.0287 | 0.0311 | 0.0339 |
| AFD-NGCF | **0.0735** | **0.1133** | **0.1869** | **0.0207** | **0.0166** | **0.0115** | **0.0511** | **0.0634** | **0.0826** | **0.0304** | **0.0331** | **0.0360** |
| GCCF | 0.0898 | 0.1346 | 0.2198 | 0.0256 | 0.0200 | 0.0138 | 0.0639 | 0.0775 | 0.0998 | 0.0387 | 0.0416 | 0.0451 |
| AFD-GCCF | **0.0977** | **0.1438** | **0.2262** | **0.0277** | **0.0213** | **0.0142** | **0.0698** | **0.0839** | **0.1056** | **0.0427** | **0.0458** | **0.0492** |
| HMLET | 0.0967 | 0.1415 | 0.2222 | 0.0276 | 0.0210 | 0.0140 | 0.0699 | 0.0835 | 0.1048 | 0.0430 | 0.0460 | 0.0494 |
| AFD-HMLET | **0.1039** | **0.1513** | **0.2367** | **0.0298** | **0.0226** | **0.0149** | **0.076** | **0.0904** | **0.1127** | **0.0471** | **0.0503** | **0.0539** |
| LightGCN | 0.1015 | 0.1466 | 0.2271 | 0.0288 | 0.0217 | 0.0144 | 0.0744 | 0.0881 | 0.1095 | 0.0465 | 0.0496 | 0.0530 |
| AFD-LightGCN | **0.1078** | **0.1584** | **0.2473** | **0.0305** | **0.0234** | **0.0156** | **0.0781** | **0.0934** | **0.1168** | **0.0486** | **0.0520** | **0.0558** |

Table B4: Performance comparisons on Amazon Book

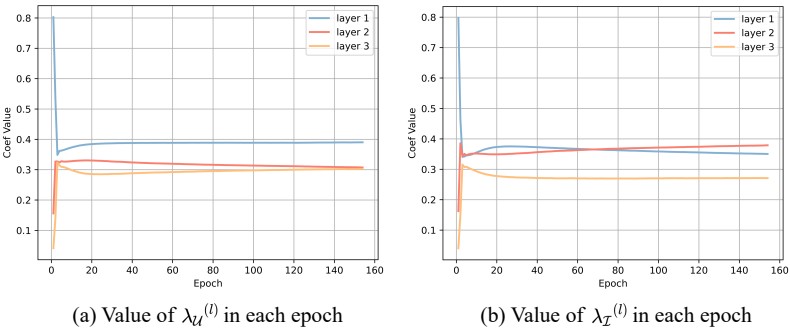

(a) Value of $\lambda_{\mathcal{U}}^{(l)}$ in each epoch

(b) Value of $\lambda_{\mathcal{I}}^{(l)}$ in each epoch

Figure B1: Penalty coefficients for user/item at each layer in each epoch during the training process (i.e., $\lambda_{\mathcal{U}}^{(l)}$, $\lambda_{\mathcal{I}}^{(l)}$) of AFD-LightGCN on Yelp dataset.

## B.3 ADDITIONAL RESULT OF HYPER-PARAMETER ANALYSIS

Figure B2 presents a hyper-parameter analysis for AFD-NGCF and AFD-GCCF. Similar to AFD-LightGCN and AFD-HMLET, the value of $\alpha$ has a certain impact on performance. An excessively large $\alpha$ can disrupt the knowledge learned by the model from historical interactions, leading to a decrease in model performance.

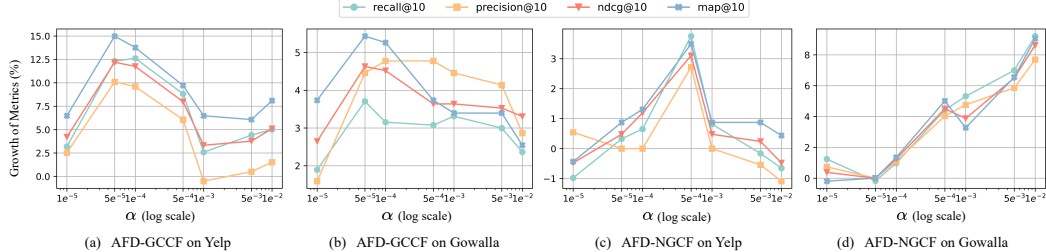

(a) AFD-GCCF on Yelp     (b) AFD-GCCF on Gowalla     (c) AFD-NGCF on Yelp     (d) AFD-NGCF on Gowalla

Figure B2: Performance comparison *w.r.t.* different $\alpha$. Using the results of the original model (*i.e.*, $\alpha = 0$) as the reference.

