# OpenReview forum: "AFDGCF: Adaptive Feature De-correlation Graph Collaborative Filtering for Recommendations"
_ICLR.cc/2024/Conference — Submitted to ICLR 2024_

### Official Review · Reviewer_TKXS · 2023-10-29

**Soundness:** 4 excellent
**Presentation:** 4 excellent
**Contribution:** 3 good
**Rating:** 8
**Confidence:** 5

**Summary:**

This paper draws attention to the challenges of over-smoothing and over-correlation in GNN-based collaborative filtering methods. In particular, the paper provides a detailed analysis of the over-correlation problem, which has been largely overlooked in existing works. Through rigorous theoretical analysis, the paper establishes a proportional association between the over-smoothing issue and the over-correlation issue, shedding light on their interconnected nature.

To tackle these issues, the paper proposes a model-agnostic constraint with adaptive weights. This constraint is designed to effectively mitigate over-smoothing and over-correlation problems in GNN-based collaborative filtering. The adaptive weights allow the constraint to dynamically adjust and optimize the learning process.

Comprehensive experiments are conducted to validate the effectiveness of the proposed constraint. The results demonstrate significant improvements in overall performance, enhanced training efficiency, and the efficacy of the adaptive approach. These findings provide strong evidence for the practical benefits of the proposed constraint in addressing the over-smoothing and over-correlation challenges in GNN-based collaborative filtering methods.

**Strengths:**

- The paper highlights the issue of decorrelation in collaborative filtering, which has received little attention in previous works.
- Through a comprehensive theoretical analysis, the paper establishes a clear association between the over-smoothing problem and the decorrelation issue.
- To address the challenges of over-smoothing and decorrelation, the paper proposes an effective solution. The proposed scheme is extensively evaluated through rigorous experiments, demonstrating its effectiveness.
- The paper is well-written and provides clear explanations. It includes illustrative figures and pilot experiments that enhance understanding and readability.

**Weaknesses:**

- I have reservations regarding the dataset preprocessing approach employed in the paper. The authors chose to exclude users and items with fewer than 15/10 interactions in some datasets. However, in my experience, this approach has the potential to create highly dense datasets and introduce bias.
- It would have been beneficial if the paper had explored the recent advancements in self-supervised learning for collaborative filtering, as these techniques have demonstrated superior performance in related studies.

**Questions:**

I would expect the authors to clarify the two issues mentioned in the weaknesses part.

---

> ### Author Response · Authors · 2023-11-17
> **Response to Reviewer TKXS**
>
> Thank you very much for carefully reviewing our paper and providing valuable questions and suggestions. Regarding your inquiries, we would like to address them as follows:
>
> * **Answer to question 1:** Thanks for your suggestion. In fact, excluding users and items with fewer than 15/10 interactions is a widely adopted practice in the evaluation of collaborative filtering tasks [1,2,3,4,5]. Even after the exclusion process, our dataset still maintains a considerable level of sparsity, as evident from the density in the table below.
>
> |  Datasets   | User # | Item # | Interaction # | Density |
> | :---------: | :----: | :----: | :-----------: | :-----: |
> |  Movielens  | 6,040  | 3,629  |    836,478    | 3.8e-2  |
> |    Yelp     | 26,752 | 19,857 |   1,001,901   | 1.8e-3  |
> |   Gowalla   | 29,859 | 40,989 |   1,027,464   | 8.4e-4  |
> | Amazon-book | 58,145 | 58,052 |   2,517,437   | 7.5e-4  |
>
> * **Answer to question 2:** Thank you very much for your suggestions. Below, we provide additional performance comparisons related to representative self-supervised method SGL [5], and we plan to supplement this baseline in the subsequent version of the paper.
>
> Additional performance comparisons of SGL on Movielens Dataset:
>
> |    Model     | Recall@10 | Precision@10 | NDCG@10 | MAP@10 |
> | :----------: | :-------: | :----------: | :-----: | :----: |
> |     DGCF     |  0.1819   |    0.1843    | 0.2477  | 0.1429 |
> |     SGL      |  0.1892   |    0.1901    | 0.2567  | 0.1496 |
> |   LightGCN   |  0.1886   |    0.1873    | 0.2540  | 0.1470 |
> | AFD-LightGCN |  0.1985   |    0.1969    | 0.2689  | 0.1594 |
>
> Thank you once again for your valuable feedback and suggestions. We sincerely hope that our responses address some of your concerns. Should you have any further questions or suggestions, please feel free to share them with us. Your input is highly appreciated.
>
>
>
> [1] He, X., Deng, K., Wang, X., Li, Y., Zhang, Y., & Wang, M. (2020, July). Lightgcn: Simplifying and powering graph convolution network for recommendation. In Proceedings of the 43rd International ACM SIGIR conference on research and development in Information Retrieval (pp. 639-648).
>
> [2] Wang, X., Jin, H., Zhang, A., He, X., Xu, T., & Chua, T. S. (2020, July). Disentangled graph collaborative filtering. In Proceedings of the 43rd international ACM SIGIR conference on research and development in information retrieval (pp. 1001-1010).
>
> [3] Wang, X., He, X., Wang, M., Feng, F., & Chua, T. S. (2019, July). Neural graph collaborative filtering. In Proceedings of the 42nd international ACM SIGIR conference on Research and development in Information Retrieval (pp. 165-174).
>
> [4] Chen, L., Wu, L., Hong, R., Zhang, K., & Wang, M. (2020, April). Revisiting graph based collaborative filtering: A linear residual graph convolutional network approach. In Proceedings of the AAAI conference on artificial intelligence (Vol. 34, No. 01, pp. 27-34).
>
> [5] Wu, J., Wang, X., Feng, F., He, X., Chen, L., Lian, J., & Xie, X. (2021, July). Self-supervised graph learning for recommendation. In Proceedings of the 44th international ACM SIGIR conference on research and development in information retrieval (pp. 726-735).

---

> > ### Comment · Reviewer_TKXS · 2023-11-23
> >
> > Thank you for your comprehensive response. I believe that the incorporated supplementary experiments would greatly enrich the evaluation section of this paper. Additionally, I suggest considering the inclusion of several additional self-supervised recommendation methods in future versions.

---

### Official Review · Reviewer_vpbW · 2023-10-31

**Soundness:** 3 good
**Presentation:** 2 fair
**Contribution:** 2 fair
**Rating:** 5
**Confidence:** 4

**Summary:**

In this paper, the authors focus on analyzing feature over-correlation in graph-based collaborative filtering, and propose an adaptive feature de-correlation regularization in graph-based collaborative filtering. Column-wise feature over-correlation will introduce redundant information for representation learning, the proposed feature de-correlation regularization can significantly improve the representation quality. Besides, the proposed feature de-correlation is very flexible and lightweight, which can coupled with representation-based CF. Experiments on several benchmarks show the effectiveness of the proposed method.

**Strengths:**

1. Interesting research topic of this paper, tacking feature over-correlation in collaborative filtering is an effective direction.
2. The proposed feature de-correlation regularization is flexible and effective in graph-based collaborative filtering. De-correlation is helpful in learning more high-quality representation for collaborative filtering.
3. Experiments conducted on several graph-based backbones demonstrate the effectiveness of the proposed de-correlation regularization.

**Weaknesses:**

1. The motivation of this paper should be highlighted. Why do the authors analyze over-correlation combined with over-smoothing? Does feature over-correlation only occur on graph-based collaborative filtering non other methods such as Matrix Factorization?
2. The reason for existing over-correlation in low-dimensional collaborative filtering is not clear. It will be more interesting if the authors deeply explain the behind reasons. Besides, does alleviating over-correlation can help to reduce over-smoothing issues in graph-based collaborative filtering? The authors should give a more explanatory illustration.
3. Lacking comparisons of related works, disentangled collaborative filtering should be involved. Besides, column-wise de-correlation can be also viewed as self-supervised learning[1]. The authors should discuss with current self-supervised graph collaborative filtering method[2,3,4].
[1]Wang X, Jin H, Zhang A, et al. Disentangled graph collaborative filtering[C]//Proceedings of the 43rd international ACM SIGIR conference on research and development in information retrieval. 2020: 1001-1010.
[2]Wu J, Wang X, Feng F, et al. Self-supervised graph learning for recommendation[C]//Proceedings of the 44th international ACM SIGIR conference on research and development in information retrieval. 2021: 726-735.
[3]Yu J, Yin H, Xia X, et al. Are graph augmentations necessary? simple graph contrastive learning for recommendation[C]//Proceedings of the 45th international ACM SIGIR conference on research and development in information retrieval. 2022: 1294-1303.
[4]Yang, Y., Wu, Z., Wu, L., Zhang, K., Hong, R., Zhang, Z., ... & Wang, M. (2023). Generative-Contrastive Graph Learning for Recommendation.

**Questions:**

Mentioned as the weakness.

---

> ### Author Response · Authors · 2023-11-17
> **Response to Reviewer vpbW**
>
> Thank you very much for carefully reviewing our paper and providing valuable questions and suggestions. Regarding your inquiries, we would like to address them as follows:
>
> * **Answer to question 1:** The reason for simultaneously investigating over-smoothing and over-correlation is rooted in their significance as crucial issues regarding the performance of GNN-based collaborative filtering. Over-smoothing has been widely acknowledged in prior research as a primary factor affecting the performance of GNN-based collaborative filtering. By contrasting it with over-correlation, we aim to highlight the latter as another equally important factor. We attribute the occurrence of over-correlation primarily to the message-passing operations in GNNs, with the degree of over-correlation increasing with the number of message-passing iterations. As for the over-correlation issue in other methods, such as MF, we have not observed similar issues. These methods lack explicit operations, like message-passing, that significantly increase redundancy in feature representations.
>
> * **Answer to question 2:** The presence of over-correlation in low-dimensional collaborative filtering is primarily ascribed to the message-passing operations inherent in GNNs. The degree of over-correlation tends to escalate with an increase in the number of message-passing iterations. In fact, the method we proposed not only addresses over-correlation but also partially alleviates over-smoothing. Due to space constraints, we did not include this aspect in Figure 5. We have supplemented the SMV below, corresponding to Figure 5.
>
> The SMV metric corresponding to Figure 5 illustrates that addressing over-correlation can help to mitigate over-smoothing issues in graph-based collaborative filtering.
>
> ![](https://z1.ax1x.com/2023/11/15/piYt0yt.png)
>
> * **Answer to question 3:** Thanks for your suggestions. Below, we provide additional performance comparisons related to DGCF [1] and representative self-supervised method SGL [2], and we plan to supplement these baselines in the subsequent version of the paper.
>
> Additional performance comparisons of DGCF & SGL on Movielens Dataset:
>
> | Model        | Recall@10 | Precision@10 | NDCG@10 | MAP@10 |
> | ------------ | --------- | ------------ | ------- | ------ |
> | DGCF         | 0.1819    | 0.1843       | 0.2477  | 0.1429 |
> | SGL          | 0.1892    | 0.1901       | 0.2567  | 0.1496 |
> | LightGCN     | 0.1886    | 0.1873       | 0.2540  | 0.1470 |
> | AFD-LightGCN | 0.1985    | 0.1969       | 0.2689  | 0.1594 |
>
> Thank you once again for your valuable feedback and suggestions. We sincerely hope that our responses address some of your concerns and contribute to a more positive perception of our work. Should you have any further questions or suggestions, please feel free to share them with us. Your input is highly appreciated.
>
>
>
> [1] Wang, X., Jin, H., Zhang, A., He, X., Xu, T., & Chua, T. S. (2020, July). Disentangled graph collaborative filtering. In Proceedings of the 43rd international ACM SIGIR conference on research and development in information retrieval (pp. 1001-1010).
>
> [2] Wu, J., Wang, X., Feng, F., He, X., Chen, L., Lian, J., & Xie, X. (2021, July). Self-supervised graph learning for recommendation. In Proceedings of the 44th international ACM SIGIR conference on research and development in information retrieval (pp. 726-735).

---

> > ### Comment · Reviewer_vpbW · 2023-11-21
> > **Resposes to authors**
> >
> > After reading the author's responses, I also have some questions:
> > (1) The major is the effectiveness of the proposed AFDGCF compared with other SOTA methods. Graph self-supervised methods are current SOTAs, I think the comparisons with SOTAs are insufficient. First, SGL is proposed in SIGIR 2021, and more advanced self-supervised methods have been proposed in recent years, such as SimGCL and more. Besides, comparisons with SGL were only conducted on Movielens-1M, which is not convincing. Movielens-1M is not suitable for conducting self-supervised experiments as it has too-high interaction sparsity(over 1% while most recommendation scenes are less than 0.01%). I think the authors should select more effective self-supervised methods and more general datasets for comparisons.  Nevertheless, I still recognize the author's contribution, to analyzing graph collaborative filtering~(GCF) from the perspective of column-wise de-correlation, which is effective in improving GCF backbones, such as LightGCN. Maybe combining self-supervised GCL methods with column-wise de-correlation is interesting. Looking forward to the authors' responses.
> > (2) I stick to my point of view that column correlation also exists in matrix factorization. Why not add experiments that set LightGCN to 0 layer, which degenerates to BPR-MF? In my view, column correlation not only exists in GNN-based CF. So, it lacks explanations of the above.

---

> > > ### Author Response · Authors · 2023-11-23
> > > **Response to Reviewer**
> > >
> > > Thank you for taking the time to read our rebuttal and raising your questions. We appreciate your input and below are our responses to your concerns:
> > >
> > > - **Response to question 1:** Thank you for your question. In this paper, our main focus is on analyzing the over-correlation issue present in general GNN-based CF models. CF models based on self-supervised learning are not the focus of our current study. However, your mention of "Combining self-supervised GCL methods with column-wise de-correlation" is indeed interesting and insightful. We will consider incorporating this into our future research endeavors. Based on your previous suggestion, we have demonstrated a comparison of SGL on the Movielens-1M dataset. Due to time constraints, we will aim to include experiments with additional datasets and more recent graph self-supervised learning methods in the future.
> > >
> > > - **Response to question 2:** From a theoretical standpoint, the correlation in Matrix Factorizations (MF) could only arise from factors such as dimension redundancy and overfitting during the training process [1,2,3]. In the early work on MF, controlling the size of dimensions, applying the Frobenius norm, and imposing a zero-mean spherical Gaussian prior are methods that have been found to effectively limit this. Therefore, we believe that the issue of over-correlation does not exist in MF. On the other hand, GNN-based CF models inherently possess correlation in representations due to message passing and aggregation operations, and this correlation persists despite well-conducted training, leading to over-correlation issues. In order to answer your question, we experimented with applying a penalty to the column-wise correlation of features during the training process of MF. We found that the results were quite distinct from those observed in GNN-based CF models, in that we did not observe a significant variation in performance under different levels of feature correlation.
> > >
> > > [1] Mnih, A., & Salakhutdinov, R. R. (2007). Probabilistic matrix factorization. Advances in neural information processing systems, 20.
> > >
> > > [2] Salakhutdinov, R., & Mnih, A. (2008, July). Bayesian probabilistic matrix factorization using Markov chain Monte Carlo. In *Proceedings of the 25th international conference on Machine learning* (pp. 880-887).
> > >
> > > [3] Shan, H., & Banerjee, A. (2010, December). Generalized probabilistic matrix factorizations for collaborative filtering. In *2010 IEEE international conference on data mining* (pp. 1025-1030). IEEE.

---

### Official Review · Reviewer_HBXd · 2023-11-04

**Soundness:** 3 good
**Presentation:** 3 good
**Contribution:** 2 fair
**Rating:** 5
**Confidence:** 4

**Summary:**

The paper analyzes the feature correlation issues in graph collaborative filtering. The author(s) present empirical studies on the smoothness and correlation of each layer of various graph collaborative filtering methods. Then, the author(s) propose AFDGCF that incorporates an auxiliary loss function to explicitly optimize the over-correlation issue. Extensive experiments on four public datasets and four popular GCF backbones show the effectiveness of the proposed method. Code is available and the author(s) promise to release all the code after the reviewing phase.

**Strengths:**

1. The paper studies an important task, i.e., graph collaborative filtering.
2. The proposed model is implemented by an open-source framework, making it easy to reproduce. Code is available during the reviewing phase.
3. Extensive experiments on four public datasets and four popular GCF backbones show the effectiveness of the proposed method.

**Weaknesses:**

1. Limited novelty. The paper seems like a straightforward application of existing literature, specifically the DeCorr [1] that focuses on general deep graph neural networks, in a specific application domain. The contribution of this study is mainly the transposition of DeCorr's insights into graph collaborative filtering, with different datasets and backbones. Although modifications like different penalty coefficients for users and items are also proposed, the whole paper still lack enough insights about what are unique challenges of overcorrelation in recommender systems.

2. It could be better if one additional figure could be illustrated, i.e., how Corr and SMV metrics evolve with the application of additional network layers—mirroring the Figure 2, but explicitly showcasing the effects of the proposed method—the authors could convincingly validate their auxiliary loss function's efficacy.

3. Presentation issues. The y-axis labels of Figure 2 lack standardization, e.g., 0.26 vs. 0.260 vs. 2600 vs. .2600.

[1] Jin et al. Feature overcorrelation in deep graph neural networks: A new perspective. KDD 2022.

**Questions:**

According to Theorem 1, there exists a proportional relationship between column correlation and row correlation of a matrix. So whether existing works on alleviating row correlation issues like contrastive learning also solve the correlation issues? Once the row correlation is alleviated, according to the proportional relationship, the column correlation should be alleviated as well. If so, why do we need the proposed auxiliary loss to explicitly alleviate the column correlation issue?

---

> ### Author Response · Authors · 2023-11-17
> **Response to Reviewer HBXd**
>
> Thank you very much for reviewing our paper. Regarding your inquiries, we would like to address them as follows:
>
> * **Reply to weakness 1:** It is important to note that we are pioneers in addressing feature over-correlation within the realm of recommendation systems. Diverging from prior research primarily focused on deep GNNs, our attention in the recommendation domain is how to integrate the advantages of GNN into collaborative filtering. Simultaneously, we delved into theoretical analyses of over-correlation and over-smoothing issues, establishing their interconnections. Conducting experimental analyses specifically targeting the over-correlation issue in GNN-based collaborative filtering models, we leveraged the unique characteristics of recommendation systems and introduced the Adaptive Feature Decorrelation method.
> * **Reply to weakness 2:** Thank you very much for your valuable suggestions. We will consider incorporating such a figure in subsequent version. Indeed, Figure 5 illustrates the Corr and performance with the application of additional network layers. Unfortunately, due to space constraints, the SMV was not presented in Figure 5. We have supplemented the SMV below, corresponding to Figure 5. This figure illustrates the variations in SMV among GNN-based collaborative filtering models with different numbers of layers, before and after the implementation of our AFD loss.
>
> The SMV metric corresponding to Figure 5:
>
> ![](https://z1.ax1x.com/2023/11/15/piYt0yt.png)
>
> * **Reply to weakness 3:** Thank you very much for your thoughtful suggestions. We will make adjustments to the y-axis labels in the subsequent version.
> * **Answer to question 1:** We appreciate your insightful observations. While we acknowledge that uniformity in contrastive learning can partially alleviate the over-smoothing issue, it only indirectly affects the correlation between features and cannot effectively address the issue of  over-correlation. In contrast, our approach directly alleviates the over-correlation problem, providing a more controllable solution. In practice, the impact on performance may not be entirely proportional between these two problems. To address your concerns more comprehensively, we provide additional performance comparisons related to representative self-supervised method SGL [1].
>
> Additional performance comparisons of SGL on Movielens Dataset:
>
> |    Model     | Recall@10 | Precision@10 | NDCG@10 | MAP@10 |
> | :----------: | :-------: | :----------: | :-----: | :----: |
> |     DGCF     |  0.1819   |    0.1843    | 0.2477  | 0.1429 |
> |     SGL      |  0.1892   |    0.1901    | 0.2567  | 0.1496 |
> |   LightGCN   |  0.1886   |    0.1873    | 0.2540  | 0.1470 |
> | AFD-LightGCN |  0.1985   |    0.1969    | 0.2689  | 0.1594 |
>
> Thank you once again for your valuable feedback and suggestions. We sincerely hope that our responses address some of your concerns and contribute to a more positive perception of our work. Should you have any further questions or suggestions, please feel free to share them with us. Your input is highly appreciated.
>
>
>
> [1] Wu, J., Wang, X., Feng, F., He, X., Chen, L., Lian, J., & Xie, X. (2021, July). Self-supervised graph learning for recommendation. In Proceedings of the 44th international ACM SIGIR conference on research and development in information retrieval (pp. 726-735).

---

> > ### Comment · Reviewer_HBXd · 2023-11-22
> > **Official Comment by Reviewer HBXd**
> >
> > Thank you for your comprehensive and thoughtful responses. I appreciate the time and effort you put into addressing the specific points raised. I would like to maintain my evaluation regarding the novelty aspect. I still perceive the overall approach as a somewhat straightforward application of de-correlation techniques to a new domain (i.e., deep general GNN -> GNN for collaborative filtering).

---

### Official Review · Reviewer_Xkxj · 2023-11-06

**Soundness:** 3 good
**Presentation:** 4 excellent
**Contribution:** 3 good
**Rating:** 8
**Confidence:** 5

**Summary:**

The paper discusses the possible connections between over-smoothing and over-correlation in graph neural networks-based recommender systems. Indeed, while over-smoothing has been debated in graph-based recommendation for quite some time now, the authors claim over-correlation is still not properly analysed as happening in graph representation learning. Through an initial empirical study, the authors demonstrate that the negative effects of the two issues seem to be directly dependent and go along with the performance degradation of the models (i.e., usually after the third message-passing layer). After that, the paper underlines how over-smoothing and over-correlation may present a direct mapping to rows and columns in the node embedding matrix, respectively, and mathematically proves that the two are proportional. In this respect, as alleviating one of the two would tackle also the other, the authors propose a loss function named adaptive feature decorrelation, that comes into a static and dynamic version. An extensive experimental setting comprising four recommendation datasets and nine baselines demonstrates the efficacy of the proposed approach. Indeed, when applied to existing graph-based recommender systems, the adaptive feature decorrelation loss function is beneficial to improve the performance in terms of recommendation accuracy and requiring much less epochs to reach convergence. Finally, an ablation study justifies the soundness of the proposed architectural choices.

**Strengths:**

+ The addressed problem (i.e., over-smoothing and over-correlation in graph-based recommendation) is relatively new to the literature.
+ The empirical analysis supported by the mathematical proofs help justifying the existing problem and opening to possible solutions.
+ The experimental setting is extensive with numerous evaluation dimensions.
+ The code and datasets are released at review time.

**Weaknesses:**

- Some details about the introduced methodology need to be clarified.
- The authors may have not considered other graph-based recommendation baselines whose solutions are like the proposed one.

**After the rebuttal.** The rebuttal clarified all weaknesses.

**Questions:**

* To the best of my understanding, I cannot find the reason why the authors state that “it is crucial to maintain the smoothness of deep representations while restricting the feature correlations of the model’s representations” (beginning of page 7). The paper seems to claim that when reducing over-correlation for deeper representations, also over-smoothing will be tackled. In this sense, I cannot see the point in the quoted statement. Would you please elaborate on that?
* Did the authors consider graph-based recommendation approaches which leverage decorrelation in a similar manner to the proposed one (e.g., disentangled graph collaborative filtering, DGCF [1]). In authors’ opinion, what would it be (even intuitively) the effect of performing a double decorrelation if the proposed loss function was applied to DGCF? Would it have a positive or a negative impact, and why?

[1] Xiang Wang, Hongye Jin, An Zhang, Xiangnan He, Tong Xu, Tat-Seng Chua: Disentangled Graph Collaborative Filtering. SIGIR 2020: 1001-1010

**After the rebuttal.** The rebuttal answered all questions.

---

> ### Author Response · Authors · 2023-11-17
> **Response to Reviewer Xkxj**
>
> Thank you very much for carefully reviewing our paper and providing valuable questions and suggestions. Regarding your inquiries, we would like to address them as follows:
>
> * **Answer to question 1:** Our decision to preserve the smoothness of deep representations while constraining the feature correlations in the model's representations is grounded in prior research highlighting the significance of embedding smoothness in the effectiveness of GNN-based RS [1,2,3]. This is because collaborative filtering methods rely on the similarity of user and item embeddings. Consequently, heedlessly diminishing representation smoothness, even with the aim of mitigating over-smoothing, is not wholly advantageous for the recommendation model's performance. Our approach aims to strike a balance between alleviating over-smoothing and embedding smoothness, while addressing the issue of feature correlations, seeking an optimal point to achieve the best recommendation performance. The advantages of doing so were demonstrated in our ablation experiments (Table 3).
> * **Answer to question 2:** Thanks for your suggestion. While both our approach and DGCF [4] employ decorrelation techniques, the motivation and modeling process of decorrelation differ between the two. In the case of DGCF, it partitions the representation matrix into several chunks, treating them as different intents. It combines this with intent-aware routing to learn disentangled embeddings, aiming to ensure relative independence among learned intents. However, within each intent, certain columns remain highly correlated. In contrast, our method focuses on decorrelating any two columns, aiming to address the issue of over-correlation through constraints. Furthermore, examining Figure 4 in the DGCF paper, we observed that finer-grained disentangling would lead to a performance decline in DGCF models. We speculate that the design of intent-aware routing in DGCF introduces a higher risk of overfitting, thus limiting its performance in finer-grained situations. Below, we provide additional performance comparisons related to DGCF, and we plan to supplement this baseline in the subsequent version of the paper. The additional performance comparisons demonstrate that our approach outperforms DGCF.
>
> Additional performance comparisons of DGCF on Movielens Dataset:
>
> |    Model     | Recall@10 | Precision@10 | NDCG@10 | MAP@10 |
> | :----------: | :-------: | :----------: | :-----: | :----: |
> |     DGCF     |  0.1819   |    0.1843    | 0.2477  | 0.1429 |
> |     SGL      |  0.1892   |    0.1901    | 0.2567  | 0.1496 |
> |   LightGCN   |  0.1886   |    0.1873    | 0.2540  | 0.1470 |
> | AFD-LightGCN |  0.1985   |    0.1969    | 0.2689  | 0.1594 |
>
> Thank you once again for your valuable feedback and suggestions. We sincerely hope that our responses address some of your concerns. Should you have any further questions or suggestions, please feel free to share them with us. Your input is highly appreciated.
>
>
>
> [1] He, X., Deng, K., Wang, X., Li, Y., Zhang, Y., & Wang, M. (2020, July). Lightgcn: Simplifying and powering graph convolution network for recommendation. In Proceedings of the 43rd International ACM SIGIR conference on research and development in Information Retrieval (pp. 639-648).
>
> [2] Zhu, T., Sun, L., & Chen, G. (2021). Graph-based embedding smoothing for sequential recommendation. IEEE Transactions on Knowledge and Data Engineering, 35(1), 496-508.
>
> [3] Kikuta, D., Suzumura, T., Rahman, M. M., Hirate, Y., Abrol, S., Kondapaka, M., ... & Loyola, P. (2022). KQGC: Knowledge Graph Embedding with Smoothing Effects of Graph Convolutions for Recommendation. arXiv preprint arXiv:2205.12102.
>
> [4] Wang, X., Jin, H., Zhang, A., He, X., Xu, T., & Chua, T. S. (2020, July). Disentangled graph collaborative filtering. In Proceedings of the 43rd international ACM SIGIR conference on research and development in information retrieval (pp. 1001-1010).

---

> > ### Comment · Reviewer_Xkxj · 2023-11-20
> >
> > Dear Authors,
> >
> > thank you for your rebuttal. In the following, I'll answer to the two points discussed in my review and addressed by your rebuttal.
> >
> > - Your answer makes sense (along with the proposed references) and sufficiently addresses my concern.
> > - Regarding the joint adoption of DGCF and your framework, your answer is quite aligned with what I originally thought. Thank you also for the additional results, which further confirm the results you presented in the original paper.

---

### Meta-Review · Area_Chair_eAx6 · 2023-12-14

**Metareview:**

The paper studies over-correlation in GNN-based collaborative filtering methods, and its connection with over-smoothing. The authors propose AFDGCF, an adaptive feature de-correlation method to tackle the issue. Extensive experiments on four public datasets and four popular GCF backbones show the effectiveness of the proposed method.

Strength: reviewers find the research problem interesting and important, and the method effective, as supported by extensive empirical results.
Weakness: limited novelty and missing comparison to related work and SOTA. Even though the authors provided more experiments and comparison to related work on disentangled graph collaborative filtering and self-supervised GNNs during the rebuttal phase,  the reviewers remain split on the scores for the paper. Given the perspectives on graph propagation increases feature correlation and the connection between feature over-correlation and over-smoothing, as well as the methods to mitigate largely overlap with a recent publication [1], we cannot accept the paper without a proper comparison to the work.

[1] Liu H, Han H, Jin W, Liu X, Liu H. Enhancing Graph Representations Learning with Decorrelated Propagation. InProceedings of the 29th ACM SIGKDD Conference on Knowledge Discovery and Data Mining 2023 Aug 6 (pp. 1466-1476).

**Justification For Why Not Higher Score:**

Reviewers raised concerns on the limited novelty. The paper applies existing literature, e.g., DeCorr that focuses on general deep graph neural networks, in a specific application domain. Reviewers also asked for more comparison to related works and SOTA. The added experiments during rebuttal was not able to fully convince reviewers otherwise. The perspectives on graph propagation increases feature correlation and the connection between feature over-correlation and over-smoothing, as well as the methods to mitigate largely overlap with a recent publication [1], which the author was not able to properly cite and compare to in the draft.

**Justification For Why Not Lower Score:**

N/A

---

### Decision · Program_Chairs · 2024-01-16

Reject